# POINT PROMPTING: COUNTERFACTUAL TRACKING WITH VIDEO DIFFUSION MODELS

**Ayush Shrivastava[1]\*, Sanyam Mehta[1]\*, Daniel Geng[1], Andrew Owens[1,2]**
[1]University of Michigan, [2]Cornell University
https://point-prompting.github.io

## ABSTRACT

Trackers and video generators solve closely related problems: the former analyze motion, while the latter synthesize it. We show that this connection enables pretrained video diffusion models to perform zero-shot point tracking by simply prompting them to visually mark points as they move over time. We place a distinctively colored marker at the query point, then regenerate the rest of the video from an intermediate noise level. This propagates the marker across frames, tracing the point's trajectory. To ensure that the marker remains visible in this counterfactual generation, despite such markers being unlikely in natural videos, we use the unedited initial frame as a negative prompt. Through experiments with multiple image-conditioned video diffusion models, we find that these "emergent" tracks outperform those of prior zero-shot methods and persist through occlusions, often obtaining performance that is competitive with specialized self-supervised models. Finally, we show that trajectories produced by pretrained generators can be distilled into a fast tracker with similar performance, serving as effective supervision for a tracking model.

## 1 INTRODUCTION

Recent generative models have shown the remarkable ability to produce temporally consistent videos. The objects within them persist across frames, through occlusion, and despite variations in camera pose and lighting. These capabilities are closely related to the *visual tracking* problem. While generation deals with producing videos that contain temporally persistent objects, tracking deals with analyzing such videos to estimate motion. A variety of methods have exploited the connections between these two problems, such as by using trackers to supervise or control video generators (Chefer et al., 2025; Burgert et al., 2025; Geng et al., 2025; Hao et al., 2018; Ardino et al., 2021) and to evaluate the temporal consistency of generated videos by measuring how "trackable" they are (Allen et al., 2025; Lai et al., 2018; Ceylan et al., 2023; Geyer et al., 2023).

In this paper, we ask whether tracking capabilities *emerge automatically* in video diffusion models, as a consequence of the close connection between the two problems. Unlike high-level understanding tasks that are naturally described by captions, like object recognition, tracking cannot easily be induced by text prompting. To elicit these capabilities from a video generator, we propose a novel approach to *counterfactual modeling* that allows us to directly obtain high-quality point tracks "zero shot" from pretrained image-conditioned video diffusion models. We simply mark the position of the query point in the initial video frame using a distinctively colored dot (Fig. 1), then propagate it to future video frames by regenerating the video using SDEdit (Meng et al., 2021). After generation, the query point's position can be estimated in each frame by basic image processing.

In counterfactual modeling (Bear et al., 2023), one carefully perturbs the input variables, then analyzes how the generation changes in response. Yet large generative models have strong priors that sometimes conflict with this goal. The marker in Fig. 1, for example, may be unnatural in some environments, and so samples from a generative model may ignore it. We use a simple but effective method to address this issue: when sampling from the model, we use the unmodified initial input frame as a negative prompt for the diffusion model, thereby guiding the model toward samples that contain the marker.

Our approach is closely related to (and takes inspiration from) a recent line of work that applies counterfactual modeling to self-supervised motion estimation (Bear et al., 2023; Venkatesh et al.,

---
\*Equal contribution.

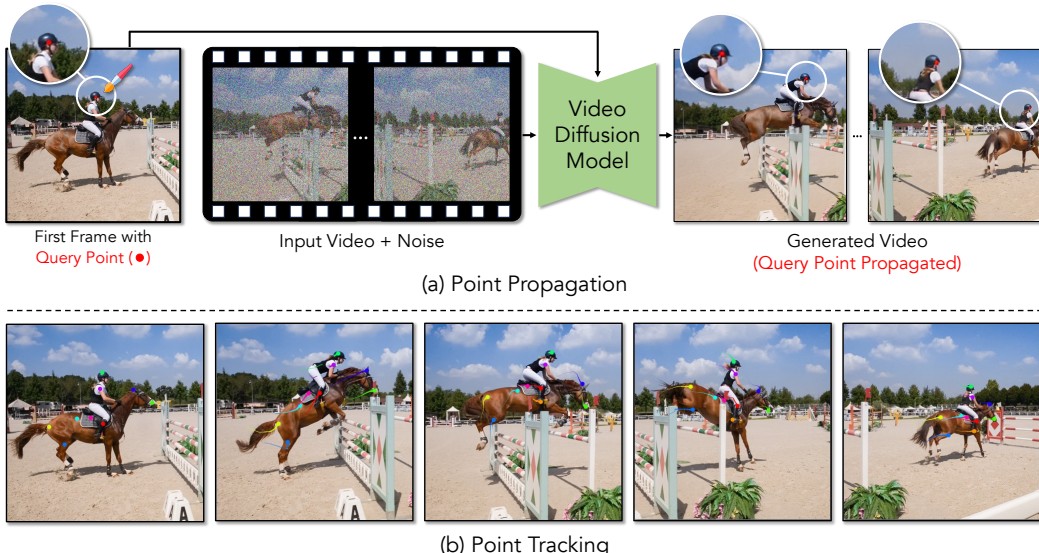

Figure 1: **Prompting a diffusion model for tracking**. (a) We use an off-the-shelf video diffusion model to perform point tracking. We add a small, distinctive marking—a red dot—to the first frame of an input video, then ask the diffusion model to regenerate the rest of the video using SDEdit (Meng et al., 2021), which propagates the marking to subsequent frames. (b) We then track the motion of this marking over time. This motion corresponds to the trajectory of the underlying physical point. The model successfully tracks through occlusion. Please see the webpage for more results: https://point-prompting.github.io.

2023). These methods train a future prediction model, then measure how the predicted future changes when a given point is perturbed in the initial frame, indicating its motion. This requires training a special-purpose model (based on masked autoencoders) that is designed specifically with this downstream use case in mind, and requires training auxiliary models to obtain high performance. By contrast, we show that *off-the-shelf* video diffusion models can track points. In this way, our work is closely related to zero-shot emergent correspondence methods (Tang et al., 2023; Zhang et al., 2023a). However, these methods treat the pretrained models as representation learners: they extract their internal features and use them to match pairs of images. Instead, we prompt a video diffusion model to visually mark point trajectories.

Our results suggest that video diffusion models are capable of tracking points through video via counterfactual modeling, without need for additional training. Through experiments on the TAP-Vid (Doersch et al., 2022) benchmark, we show:

- Pretrained video diffusion models can be directly used as visual trackers.
- The object permanence capabilities of generative models enable tracking through occlusion.
- Points can be reliably propagated through video using a novel diffusion prompting strategy.
- Tracking performance improves through iterative refinement using inpainting.
- We significantly outperform previous zero-shot tracking methods, such as those that use features from pretrained image diffusion models.
- Trajectories produced by pretrained video generators can be distilled into a fast tracker with comparable performance.

We see this work as being a step toward understanding the capabilities of large, pretrained video diffusion models, and new ways to extract these capabilities from them.

## 2 RELATED WORK

**Self-supervised Motion Estimation.** Deep learning has significantly advanced motion estimation. Early dense optical flow methods (Dosovitskiy et al., 2015; Sun et al., 2018; Teed & Deng, 2020) showed strong performance but often struggle with long-range tracking and occlusions. Inspired by Sand & Teller (2008), recent methods instead track individual points over time (Harley et al., 2022; Doersch et al., 2022), with newer architectures (Doersch et al., 2023; Karaev et al., 2024c;a; Neoral et al., 2024; Zheng et al., 2023; Doersch et al., 2024; Zholus et al., 2025) improving long-term accuracy. However, these models often rely on synthetic data, limiting their real-world generalization. To bridge this gap, self-supervised optical flow methods (Jonschkowski et al., 2020; Liu et al., 2019;

Huang et al., 2023) have been proposed, but they inherit many limitations of supervised approaches. Other work focuses directly on long-range tracking: Vondrick et al. (2018) train a model to propagate color in grayscale videos, implicitly learning motion. Cycle consistency has also been used (Jabri et al., 2020; Wang et al., 2019), including for point tracking (Shrivastava & Owens, 2024). Models trained for semantic understanding, such as DINOv2 (Oquab et al., 2023), have also been adapted for semantic and temporal correspondence. DIFT (Tang et al., 2023), based on image diffusion models, extracts features suitable for matching, while SD-DINO (Zhang et al., 2023a) combines Stable Diffusion and DINO features to solve a range of semantic and geometric tasks. A recent concurrent work (Nam et al., 2025) extracts features from a pretrained video model for tracking, using a one-to-one frame-to-latent mapping to avoid temporal compression, but involves a complex, architecture-dependent analysis to identify which layers provide the best features and does not handle occlusion. Instead of performing feature extraction, our method prompts a video diffusion model, and thus it is architecture-agnostic. It also handles occlusion by exploiting the ability of modern video diffusion models to successfully convey object permanence, which is not possible with existing methods that work by matching individual pairs of video frames.

**Counterfactual Modeling.** Prior work has explored counterfactual reasoning for visual understanding. Visual Jenga (Bhattad et al., 2025) progressively removes objects from a single image until only the background remains, revealing geometric relationships among scene elements. Recent research on counterfactual world modeling (Bear et al., 2023; Venkatesh et al., 2023) tackles keypoint prediction and optical flow by training a masked autoencoder for future-frame prediction, then perturbing inputs to estimate motion. In contrast, we exploit properties of diffusion, such as the ability to subtly manipulate videos, to obtain our predictions from an off-the-shelf model; we base our approach on generative video models rather than masked future frame prediction; and we address the long-range point tracking problem rather than optical flow. Recently, Stojanov et al. (2025) extended the counterfactual world modeling to point tracking by learning RGB perturbations that can be propagated through a frozen next-frame predictor, optimizing them with a jointly trained sparse optical-flow module. By contrast, our approach relies entirely on prompting a frozen video diffusion model and requires no additional training.

**Pretrained Models.** Large pretrained models have become foundational in computer vision, replacing task-specific architectures across classification, detection, and segmentation (Donahue et al., 2014; Chen et al., 2020; He et al., 2020; Zhang et al., 2016; Oquab et al., 2023; Radford et al., 2021; Zhai et al., 2023; Kirillov et al., 2023; Yang et al., 2024a; Liu et al., 2024; Tong et al., 2024; Li et al., 2023). Diffusion models for image generation (Podell et al., 2023; Rombach et al., 2022; Dhariwal & Nichol, 2021; Nichol et al., 2021) introduced generative features that capture semantic correspondences (Tang et al., 2023; Luo et al., 2023; Zhang et al., 2023a), but lack temporal reasoning needed for motion-centric tasks. Video diffusion models (Blattmann et al., 2023a;b; Yu et al., 2023; Wang et al., 2025; Yang et al., 2024b; Polyak et al., 2024; Chefer et al., 2025) address temporal consistency, though many still prioritize appearance over motion. Recently, Chefer et al. (2025) address this by incorporating optical flow during training. We work in the opposite direction, using generative models to aid motion estimation.

**Visual Prompting.** Prompting strategies have achieved notable success in natural language processing (Wei et al., 2022; Kojima et al., 2022), motivating analogous techniques in computer vision. One prominent direction frames downstream vision tasks as inpainting problems, using pretrained models to complete images conditioned on visual cues (Bar et al., 2022; Wang et al., 2023; Bai et al., 2024). Another line of work focuses on optimizing prompt representations, showing that both textual and visual prompts can be refined via gradient-based methods to better adapt vision models (Zhou et al., 2022; Bahng et al., 2022). Recent studies also demonstrate that simple visual prompts, such as colored shapes, can elicit useful behaviors from vision-language models (Shtedritski et al., 2023; Yao et al., 2024). We introduce a simple yet effective visual prompt: placing a colored dot at the pixel to be tracked. To our knowledge, this is the first use of image prompting for point tracking in video diffusion models.

**Controllable Generation.** Controllable generation is a key goal in generative modeling (Hao et al., 2018; Zhuang et al., 2021; Liu et al., 2021; Jo & Park, 2019; Chen et al., 2024; Zhang et al., 2023b; Ruiz et al., 2023; Chen et al., 2023). SDEdit (Meng et al., 2021) introduced a training-free method for guided synthesis using noise perturbation and iterative denoising. More recent work enables fine-grained spatial control in diffusion models (Chen et al., 2024; Lugmayr et al., 2022; Si et al., 2024; Wu et al., 2024; Chefer et al., 2023). RePaint (Lugmayr et al., 2022), for example, inpaints

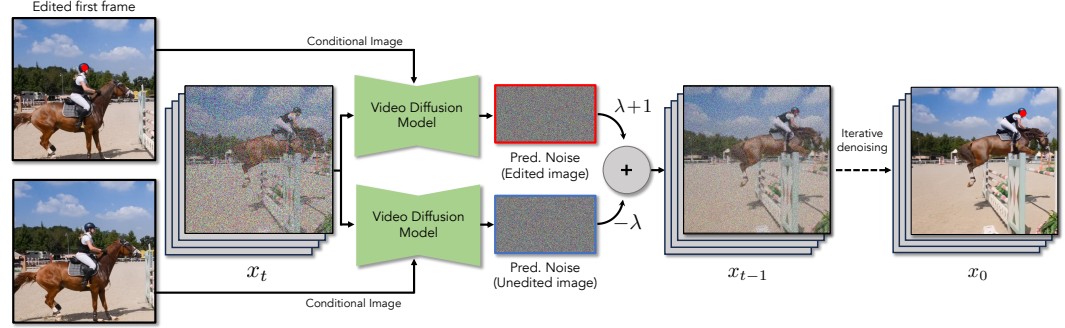

Figure 2: **Enhancing the Counterfactual Signal.** We use negative prompting to ensure that the generated video contains the marker. In each denoising step (Eq. 5), we condition the denoising on two images: (1) *Edited First Frame*: the first frame of the video with a marking added, and (2) *Unedited First Frame*: the original first frame of the video. We then subtract the weighted noise vector of the latter from the former.

masked regions without affecting the rest of the image. Methods like ControlNet (Zhang et al., 2023b) and DreamBooth (Ruiz et al., 2023) enable control via fine-tuning. These ideas have been extended to video (Zhang et al., 2023c; Feng et al., 2024), providing structured editing through architectural design and hierarchical sampling.

## 3 METHOD

Our goal is to repurpose a pretrained generative video model to track points in a video. To do this, we exploit several key properties of diffusion models. We review diffusion models, then describe how they can be adapted for point tracking.

### 3.1 PRELIMINARIES: VIDEO DIFFUSION MODELS

Latent video diffusion models (Sohl-Dickstein et al., 2015; Ho et al., 2020; Rombach et al., 2022; Blattmann et al., 2023a; Wang et al., 2025) generate a sequence of $F$ RGB frames, $\mathbf{V} \in \mathbb{R}^{F \times H \times W \times 3}$. These models operate on a compact latent representation $\mathbf{x} \in \mathbb{R}^{F' \times H' \times W' \times C}$, where $C$ is the channel dimension, which can be converted into a video via a decoder.

**Forward (Noising) Process.**[1] Given a clean video latent $\mathbf{x}_0$, we define the noising process using a variance schedule $\beta_t$ over timesteps $t \in \{1, \ldots, T\}$. The corrupted latent is constructed via:

$$\mathbf{x}_t = \sqrt{\alpha_t}\mathbf{x}_{t-1} + \sqrt{1 - \alpha_t}\boldsymbol{\epsilon}, \quad \boldsymbol{\epsilon} \sim \mathcal{N}(\mathbf{0}, \mathbf{I}), \tag{1}$$

where $\alpha_t = 1 - \beta_t$ and $\bar{\alpha}_t = \prod_{s=1}^{t} \alpha_s$.

**Reverse (Denoising) Process.** At each timestep $t$, the video diffusion model, $\boldsymbol{\epsilon}_\theta(\mathbf{x}_t, t, c)$, predicts the noise component. These models may be conditioned on additional data $c$, such as a text prompt or the desired first frame of the video. We denoise the corrupted latent (Sohl-Dickstein et al., 2015; Ho et al., 2020):

$$\mathbf{x}_{t-1} = \frac{1}{\sqrt{\alpha_t}}\left(\mathbf{x}_t - \frac{\beta_t}{\sqrt{1 - \bar{\alpha}_t}}\boldsymbol{\epsilon}_\theta(\mathbf{x}_t, t, c)\right) + \sigma_t \mathbf{z} \tag{2}$$

where $\sigma_t^2$ is the variance, and $\mathbf{z} \sim \mathcal{N}(0, I)$.

**Video Manipulation.** Trained diffusion models can also be used to manipulate existing videos, without additional training. We discuss two such applications: regeneration and inpainting.

Rather than generating a latent vector from scratch, one can regenerate an existing, clean video with modifications using SDEdit (Meng et al., 2021). We add an intermediate level of noise, $1 < t < T$:

$$\mathbf{x}_t = \sqrt{\bar{\alpha}_t}\mathbf{x}_0 + \sqrt{1 - \bar{\alpha}_t}\boldsymbol{\epsilon}, \tag{3}$$

---

[1]Our method is agnostic to the specific diffusion model and therefore follows the widely used standard notation of denoising diffusion models (Ho et al., 2020) with classifier-free guidance (Ho & Salimans, 2022).

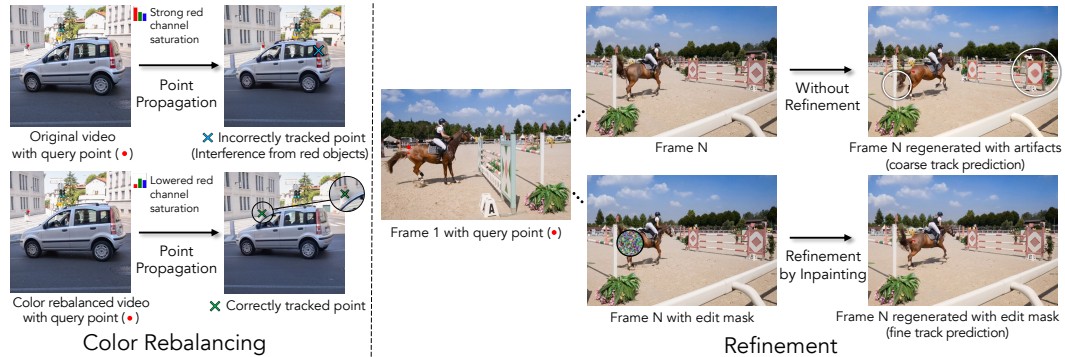

Figure 3: **Tracking Enhancements.** To improve point tracking in video, we introduce two enhancements: (1) *Color Rebalancing:* remove existing red hues to ensure the red marker remains a unique tracking cue; (2) *Refinement:* obtain initial trajectories with a color-based tracker, then refine them using an inpainting mask to correct temporal artifacts such as object shifts (as shown in white circles). This two-step procedure first produces coarse tracks and then refines them via mask-constrained reverse diffusion.

and then run the reverse diffusion process to denoise it. This results in a video that resembles the coarse structure of the original, but with different fine-grained details (e.g., restyling a real video into a cartoon using a text prompt).

We can also use pretrained video diffusion models for inpainting (Lugmayr et al., 2022). Given a binary spatiotemporal mask $\mathbf{m} \in \mathbb{B}^{F \times H \times W}$ indicating which patches of the input video can (and cannot) be changed, we run the reverse diffusion process and constrain updates to the masked region. At each denoising step, we constrain the updates such that they occur only in the masked region. In each step of the reverse diffusion process, we compute (Lugmayr et al., 2022):

$$
\begin{aligned}
\tilde{\mathbf{x}}_{t-1} &= \frac{1}{\sqrt{\alpha_t}} \left( \mathbf{x}_t - \frac{\beta_t}{\sqrt{1 - \bar{\alpha}_t}} \boldsymbol{\epsilon}_\theta \right) + \sigma_t \mathbf{z}, \quad \mathbf{z} \sim \mathcal{N}(\mathbf{0}, \mathbf{I}), \\
\mathbf{x}_{t-1}^{\text{original}} &= \sqrt{\bar{\alpha}_{t-1}} \mathbf{x}_0 + \sqrt{1 - \bar{\alpha}_{t-1}} \boldsymbol{\epsilon}, \quad \boldsymbol{\epsilon} \sim \mathcal{N}(\mathbf{0}, \mathbf{I}), \\
\mathbf{x}_{t-1} &= \mathbf{m} \odot \tilde{\mathbf{x}}_{t-1} + (1 - \mathbf{m}) \odot \mathbf{x}_{t-1}^{\text{original}},
\end{aligned}
\tag{4}
$$

where $\boldsymbol{\epsilon}_\theta$ is the estimated noise for the iteration $t$, and as before $\mathbf{x}_0$ is the latent for the input video.

## 3.2 POINT PROMPTING FOR COUNTERFACTUAL TRACKING

We now describe an approach to counterfactual modeling that enables a video diffusion model to perform "zero shot" tracking.

**Marking a Point's Trajectory.** Given an input video and the pixel location of a query point, our goal is to predict the positions of the point in the subsequent frames. As shown in Fig. 1, we prompt an off-the-shelf video diffusion model to draw a distinctive marker in each frame at the point's position. We then localize the point position using simple low-level image processing.

We insert a distinctive marking on the query point's position in the initial frame. For this, we simply use a circular dot, which can plausibly be interpreted as being part of the object's surface. For simplicity, we color this dot pure red in all of our experiments. We then apply SDEdit (Sec. 3.2) using an intermediate timestep $1 < t < T$ to the video to manipulate the video, while conditioning on the edited initial frame. This propagates the marker to the subsequent frames of the video.

**Enhancing the Counterfactual Signal.** One of the challenges of applying counterfactual modeling to powerful generative models is that their strong priors lead them to ignore the manipulations that we introduce. For example, when the marker does not naturally fit into a scene, it will often disappear from the generated video within a few frames. We address this problem by using a simple negative prompt that reduces the probability of drawing samples that resemble the original video. We compute the difference between two noise estimates (Fig. 2) that are computed using different types of first-frame conditioning: one where we condition on the original image (i.e., without the marker) and another where we condition on the edited image (i.e., with the marker):

$$
\tilde{\boldsymbol{\epsilon}}_\theta \left( \mathbf{x}_t, \mathbf{c}_I \right) = (\lambda + 1) \cdot \boldsymbol{\epsilon}_\theta(\mathbf{x}_t, \phi(\mathbf{c}_I)) - \lambda \cdot \boldsymbol{\epsilon}_\theta \left( \mathbf{x}_t, \mathbf{c}_I \right),
\tag{5}
$$

where $\tilde{\epsilon}$ is the noise estimate after enhancement, $\mathbf{c}_I$ is the initial-frame conditioning, $\phi(\mathbf{c}_I)$ is the initial frame after applying the counterfactual manipulation (i.e., adding the marker), and $\lambda > 0$ is a weight. Due to the well-known connection between denoising and score functions, the modified denoiser $\tilde{\epsilon}$ corresponds to the following score function (Ho & Salimans, 2022; Karras et al., 2024):

$$\nabla_{\mathbf{x}_t} \log(p_\lambda(\mathbf{x}_t)) = \nabla_{\mathbf{x}_t} \log \left( p(\mathbf{x}_t \mid \phi(\mathbf{c}_I)) \left[ \frac{p(\phi(\mathbf{c}_I) \mid \mathbf{x}_t)}{p(\mathbf{c}_I \mid \mathbf{x}_t)} \right]^\lambda \right), \tag{6}$$

where $p(\mathbf{x}_t)$ is the probability under the model for the noisy input at time $t$, and where we have used the well-known fact that $\epsilon(\mathbf{x}_t) \propto -\nabla_{\mathbf{x}_t} \log(p(\mathbf{x}_t))$ and Bayes rule, following the standard derivation of classifier-free guidance (Ho & Salimans, 2022). From this perspective, we see that our sampling procedure generates videos conditioned on the manipulated initial frame, while biasing the score direction away from samples from the unedited conditioning.

We note that this strategy is related to (but distinct from) the approach used in previous work on counterfactual world models (Bear et al., 2023; Stojanov et al., 2025). They generate two possible future images using a masked autoencoder: one with the marker and one without. They then enhance the signal by directly subtracting the two generated images, which amounts to approximately estimating: $\mathbb{E}_{p(\mathbf{x}|\phi(\mathbf{c}_I))}[\mathbf{x}] - \mathbb{E}_{p(\mathbf{x}|\mathbf{c}_I)}[\mathbf{x}]$. Like our approach, this method enhances their ability to detect the effect of the counterfactual by comparing the generated result to an unedited baseline, but instead of comparing the predicted samples themselves, we include this constraint as guidance in the sampler. In our experiments, we found that objects often subtly change position in different samples of a video diffusion model, leading to this differences between generations to contain significant artifacts, making it challenging to use this approach.

**Tracking the Marker.** To extract a track from generated videos containing an inserted marker at a query point, we implement a very simple tracker that locates the marker in each frame based on color. Given the marker's initial location $(u_0, v_0)$ in the first frame, we track its motion frame by frame. For each subsequent frame $k$, the tracker searches for red pixels (in HSV colorspace) within a local window of radius $r$ centered at the previous location $(u_{k-1}, v_{k-1})$, selecting the pixel closest to the previous position. Since the marker appears as a small blob, we refine the estimate by averaging the positions of nearby red pixels to obtain a more stable center, which serves as the predicted track point.

If no red pixels are found within the search region, we treat the marker as occluded and propagate the last known position forward. We expand the search radius $r$ at each step until the marker reappears, after which we reset $r$ to its original value. This adaptive strategy makes the tracker robust to temporary occlusions and large displacements, enabling it to recover from tracking uncertainty.

### 3.3 EXTENSIONS

We can further improve the prediction by coarse-to-fine refinement and by rebalancing the colors in the video to exclude the marker's color (Fig. 3).

**Coarse-to-Fine Refinement.** Accurate tracking requires that the generated video remain pixel-aligned with the original. However, the generated video may be subtly misaligned with the original video after regeneration, leading to tracking errors. Inspired by coarse-to-fine motion estimation, we improve our tracking predictions after their initial estimates, by exploiting the fact that video diffusion models can be repurposed to perform inpainting. We restrict the model's ability to modify the video during generation, allowing it to generate only regions near the potential tracked point, while preserving the rest of the video content.

After obtaining the initial estimate of marker positions (as described above), we construct a spatiotemporal binary mask $\mathbf{m} \in \mathbb{R}^{F \times H \times W}$, where each frame's mask is set to 1 within a small radius $r$ centered on the tracked location, i.e., $\mathbf{m}[u, v]$ is set to 1 if $(u, v) \in B_r(u_k, v_k)$. We then re-run the video generation, while allowing only the image regions indicated by $\mathbf{m}$ to change, using Eq. 4.

**Color Rebalancing.** Since our tracker relies on detecting a particular color, we rebalance the video's colors such that the marker's color does not appear within it. We do this by reducing the saturation of the marker's color. For example, when tracking a red marker, we reduce the saturation of red regions, effectively suppressing natural red hues while preserving overall image quality (details provided in Appendix B.1). We find that this reduces mistakes during occlusion, since the marker is not present and thus false detections are more common.

## 4 EXPERIMENTS

We evaluate our prompting strategy's ability to accurately track points through a video, using the TAP-Vid benchmarks (Doersch et al., 2022).

### 4.1 VIDEO MODELS

We consider recent image-conditioned video diffusion models:

**Wan2.1** (Wang et al., 2025) combines a 3D causal VAE with a diffusion transformer (DiT) conditioned on text and an input image and trained using flow-matching (Lipman et al., 2022). The VAE encodes video into latents $x \in \mathbb{R}^{(1+F/4) \times H/8 \times W/8}$, keeping the first frame at full temporal resolution and downsampling the rest by $4\times$. Outputs are $480 \times 832$. We test 1.3B- and 14B-parameter variants, reporting results with the 14B model unless noted.

**Wan2.2** (Wang et al., 2025) extends Wan2.1 with a Mixture-of-Experts (MoE) architecture. By distributing denoising across timesteps among specialized experts, it increases model capacity without extra computation and is trained on a much larger dataset.

**CogVideoX** (Yang et al., 2024b) is another image-to-video (I2V) diffusion model that also combines a 3D causal VAE with a diffusion transformer. It generates $768 \times 1360$ videos from a text prompt and reference image. The VAE compresssion is the same as Wan, while the transformer conditions on the image and T5 text embeddings (Raffel et al., 2020).

For all models, we use 50 denoising steps with noise strength 0.5 and an empty text prompt. Experiments run on A40 or L40S GPUs (one GPU per video). Generating a 50-frame video for a single query point takes about 7 min for Wan2.1-1.3B, 30 min for Wan2.1-14B, and 20 min for CogVideoX. These runtimes are acceptable given our focus on evaluating the tracking capabilities of video diffusion models. We note that method could potentially be distilled into a more efficient model, similar to Opt-CWM (Stojanov et al., 2025).

### 4.2 TAP-VID BENCHMARK

We evaluate on two TAP-Vid benchmark splits: DAVIS (30 videos, 34–104 frames) and Kinetics (30 sampled videos, 250 frames, following (Stojanov et al., 2025)) for efficiency. These natural videos match the training distribution of our video diffusion models (rather than computer generated video). Using the first sampling strategy, we pick one query point per video, overlay a red dot at its position in the first frame, and run our model to propagate the point throughout the video. The resulting trajectory is then extracted using our tracker.

**Evaluation Metrics.** We report: (1) *Positional Accuracy* ($\delta^x_{\text{avg}}$), fraction of visible points within distance thresholds; (2) *Occlusion Accuracy* (OA), visibility prediction accuracy; and (3) *Average Jaccard* (AJ), average overlap between predicted and ground-truth visible points across thresholds (Doersch et al., 2022).

## 5 RESULTS

Unless otherwise noted, we use Wan2.1-14B (Wang et al., 2025) as the video diffusion model for all experiments.

**Quantitative Results.** Table 1 compares our method against several baselines using Wan2.1. Among zero-shot methods, ours achieves the highest performance. On TAP-Vid DAVIS, we reach an AJ score of 42.21, outperforming all other zero-shot baselines and even surpassing GMRW (Shrivastava & Owens, 2024), a strong self-supervised approach. Our occlusion accuracy also exceeds that of both zero-shot and self-supervised methods, approaching supervised performance, highlighting the ability of diffusion models to reason through occlusions.

We include top supervised methods such as CoTracker3 (Karaev et al., 2024b) and TAPNext (Zholus et al., 2025), as well as the best-performing self-supervised baseline, Opt-CWM (Stojanov et al., 2025). While conceptually related, Opt-CWM learns to propagate perturbations through a next-frame predictor supervised by sparse flow. In contrast, our method is entirely zero-shot, using a simple colored dot without training or learned perturbations. As shown in Table 6, we also report results on TAP-Vid Kubric, where overall performance is comparatively lower, likely due to the dataset's synthetic nature and the fact that existing video models are primarily trained on real videos.

| Method | Supervision | TAP-Vid DAVIS | | | TAP-Vid Kinetics | | |
|---|---|---|---|---|---|---|---|
| | | AJ ↑ | $< \delta^x_{avg}$ ↑ | OA ↑ | AJ ↑ | $< \delta^x_{avg}$ ↑ | OA ↑ |
| RAFT (Teed & Deng, 2020) | Supervised | 34.48 | 53.55 | 74.90 | 30.15 | 46.44 | 75.44 |
| TAP-Net (Doersch et al., 2022) | | 32.05 | 48.42 | 77.35 | 34.59 | 48.42 | 80.88 |
| TAPIR (Doersch et al., 2023) | | 58.47 | 70.56 | 87.27 | 47.46 | 59.56 | 85.76 |
| CoTracker3 (Karaev et al., 2024b) | | 64.45 | 77.13 | 90.90 | **54.35** | **65.99** | **89.43** |
| TAPNext (Zholus et al., 2025) | | **66.56** | **79.48** | **92.21** | 52.97 | 64.46 | 89.30 |
| GMRW (Shrivastava & Owens, 2024) | Self-Sup. | 36.47 | 54.59 | 76.36 | 25.70 | 41.63 | 71.33 |
| Opt-CWM (Stojanov et al., 2025) | | **47.53** | **64.83** | **80.87** | **44.85** | **57.74** | **84.12** |
| DINOv2+NN (Oquab et al., 2023) | Zero-Shot | 15.19 | 31.19 | 61.81 | 12.69 | 24.22 | 62.45 |
| DIFT (Tang et al., 2023) | | 21.51 | 39.55 | 69.71 | 15.10 | 25.56 | 63.17 |
| SD-DINO (Zhang et al., 2023a) | | 29.68 | 50.45 | 69.71 | 16.47 | 28.37 | 62.79 |
| Ours | | **42.21** | **57.29** | **82.90** | **27.36** | **41.51** | **71.39** |

Table 1: **TAP-Vid Benchmark Results.** We report results on the TAP-Vid First benchmark. Our zero-shot method outperforms all other zero-shot baselines and is competitive with self-supervised and supervised trackers. On TAP-Vid DAVIS-First, it matches self-supervised methods in AJ and exceeds them in occlusion accuracy, highlighting strong object permanence from generative modeling.

| Method | TAP-Vid DAVIS | | |
|---|---|---|---|
| | AJ ↑ | $< \delta^x_{avg}$ ↑ | OA ↑ |
| CogVideoX1.5-5B (Yang et al., 2024b) | 24.15 | 34.38 | 70.79 |
| Wan2.1-1.3B (Wang et al., 2025) | 44.58 | 58.77 | 85.16 |
| Wan2.1-14B (Wang et al., 2025) | 48.60 | 63.47 | 85.75 |
| Wan2.2-14B (Wang et al., 2025) | 48.78 | 63.91 | 86.17 |

Table 2: **Video Model Ablations.** Wan2.1-1.3B and 14B (Wang et al., 2025) outperform CogVideoX (Yang et al., 2024b), showing that stronger video models improve tracking performance.

| Image source | TAP-Vid DAVIS | | |
|---|---|---|---|
| | AJ ↑ | $< \delta^x_{avg}$ ↑ | OA ↑ |
| DAVIS (256×256) | 42.21 | 57.29 | 82.90 |
| DAVIS (256×256 up.) | 45.48 | 60.16 | 83.49 |
| DAVIS (original res.) | 48.60 | 63.47 | 85.75 |

Table 3: **Image Resolution Ablations.** Comparing input resolutions for Wan2.1. Upscaling with (Zhou et al., 2024) improves tracking by better aligning with the model's training distribution.

| Method | TAP-Vid DAVIS | | |
|---|---|---|---|
| | AJ ↑ | $< \delta^x_{avg}$ ↑ | OA ↑ |
| all | 48.60 | 63.47 | 85.75 |
| w/o refinement | 42.70 | 59.26 | 85.14 |
| w/o counterfactual enhancement | 22.03 | 38.53 | 61.19 |
| w/o color rebalancing | 34.86 | 52.12 | 82.18 |
| tracker only | 11.26 | 21.07 | 77.74 |

Table 4: **Tracking Pipeline Ablations.** Quantitative results on TAP-Vid DAVIS-First showing the impact of each stage in our pipeline (Fig. 3). The last row uses original pixel color instead of the red dot for tracking.

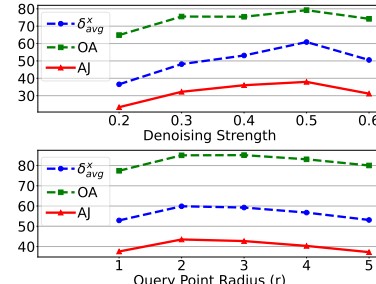

Figure 4: **Effect of denoising strength and radius on tracking performance.**

**Different Video Models.** Table 2 shows results using Wan2.1 (1.3B and 14B variants), Wan2.2, and CogVideoX (Yang et al., 2024b). Our method successfully tracks points for all four models, demonstrating compatibility across different video generation backbones. Wan2.1 and Wan2.2 obtain the strongest results, with the 14B variant outperforming the 1.3B model. We attribute this gain to their higher video generation quality, suggesting that improved generative fidelity directly may improved tracking ability.

**Generation Resolution.** The TAP-Vid benchmark provides videos at a resolution of 256×256, which we resize to 480×832 to match the input resolution of Wan2.1. To assess the impact of resolution, we first upsample inputs using Upsample-A-Video (Zhou et al., 2024), which improves tracking (Table 3). We then run Wan2.1 on the original high-res DAVIS frames (Perazzi et al., 2016), achieving an AJ score of 48.6, surpassing Opt-CWM. These results show that higher-resolution inputs significantly enhance tracking by improving video generation quality.

**Point Propagation Ablations.** Table 4 shows ablations of key components. The first row shows our full model with all components enabled. Removing the inpainting-based refinement step reduces positional accuracy due to spatial shifts during denoising which negatively affects tracking precision. Removing counterfactual enhancement guidance causes failure in point propagation where tracking

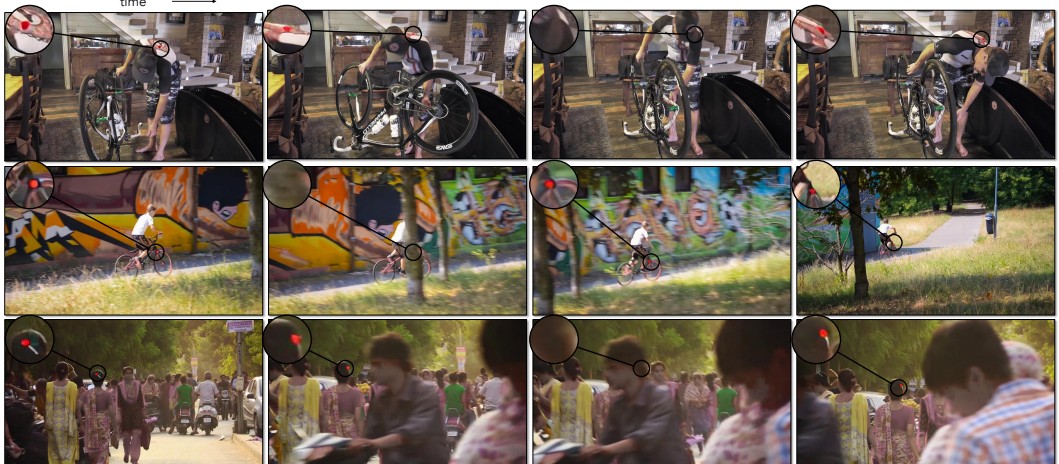

Figure 5: **Point Propagation.** Frames generated from the video diffusion model show consistent red dot tracking. The model recovers the point after long occlusions, showing temporal understanding and object permanence.

is lost after 5–6 frames, highlighting its critical role in maintaining point consistency across frames. Disabling color rebalancing also degrades performance. Since the tracker relies on detecting red pixels, failure to suppress red tones in the background introduces false positives, especially when the query point is occluded, making tracking less reliable.

We also evaluate a tracker-only baseline that tracks the query point's color from the initial frame without any point propagation. This performs significantly worse, highlighting that the primary performance gains in our method arise from accurate point propagation through video generation, rather than from the tracker itself, which is intentionally kept simple. Additionally, we ablate key hyperparameters in Fig. 4. We observe that a noise strength of 0.5 and a query point radius of 2 pixels yield the best results. The effect of varying the marker color is further analyzed in Table 9.

**Distilling the Generator into a Tracker.** While our approach shows strong zero-shot tracking performance, it requires a separate video generation for each query point, which limits efficiency. To obtain a fast tracker, inspired by this (Stojanov et al., 2025), we distill our generation-based method into an efficient tracking architecture. Specifically, we collect pseudo-label trajectories by running our marker propagation method on 1,000 unlabeled videos from the TAP-Vid Kinetics dataset. These extracted trajectories serve as supervision to train CoTracker (Karaev et al., 2024d) from scratch.

| Method | TAP-Vid DAVIS | | |
|---|---|---|---|
| | AJ $\uparrow$ | $< \delta_{\text{avg}}^x \uparrow$ | OA $\uparrow$ |
| Wan2.1-1.3B (teacher) | 38.78 | 54.75 | 85.00 |
| Cotracker (distilled) | 37.17 | 53.12 | 84.24 |

Table 5: **Distilling the generator.** We distill our method to CoTracker which achieves performance close to the teacher Wan2.1 and runs orders of magnitude faster.

Table 5 compares the teacher model used to generate trajectories with the distilled CoTracker model. Despite being trained solely on pseudo-labels produced by our zero-shot method, the distilled tracker achieves performance very close to the teacher while running orders of magnitude faster at inference. This indicates that the temporal reasoning capabilities of large video diffusion models can be transferred into a lightweight tracking network, effectively bridging generative zero-shot tracking and efficient feed-forward inference.

**Qualitative Results.** In Fig. 5, we show video generations from our method, where red dots are successfully propagated across frames, including through occlusions. We extract these points and display the resulting tracks for multiple query points in Fig. 6. Our method reliably tracks points over long temporal range and maintains accuracy even in the presence of occlusions.

## 6 LIMITATIONS

Our approach requires generating a video for each tracked point. Since our goal is to show that video generators can perform tracking, rather than to perform tracking as an end in itself, we did not

time

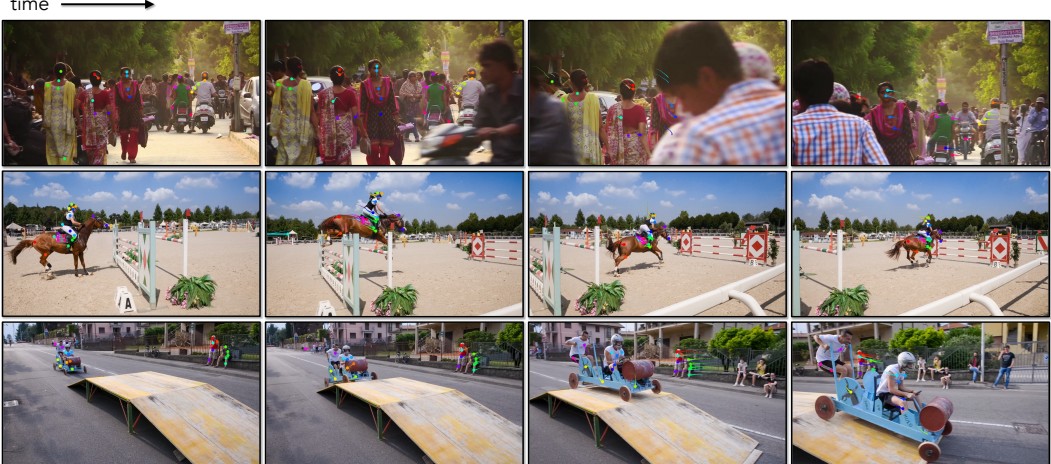

Figure 6: **Tracking results.** Frames show the query point being tracked (circled dot) and its trajectory over the previous 5 frames. When the query point is occluded, only the trajectory tail is displayed without the dot.

attempt to optimize our approach. However, it can potentially be addressed by distilling our model's predictions into a network that directly performs tracking, by considering more efficient generation methods (e.g., one-step sampling), or by tracking multiple points at once. The video generators also sometimes fail to interpret the red dot as being attached to the object surface, especially for (likely out-of-distribution) computer-generated videos (Fig. 7).

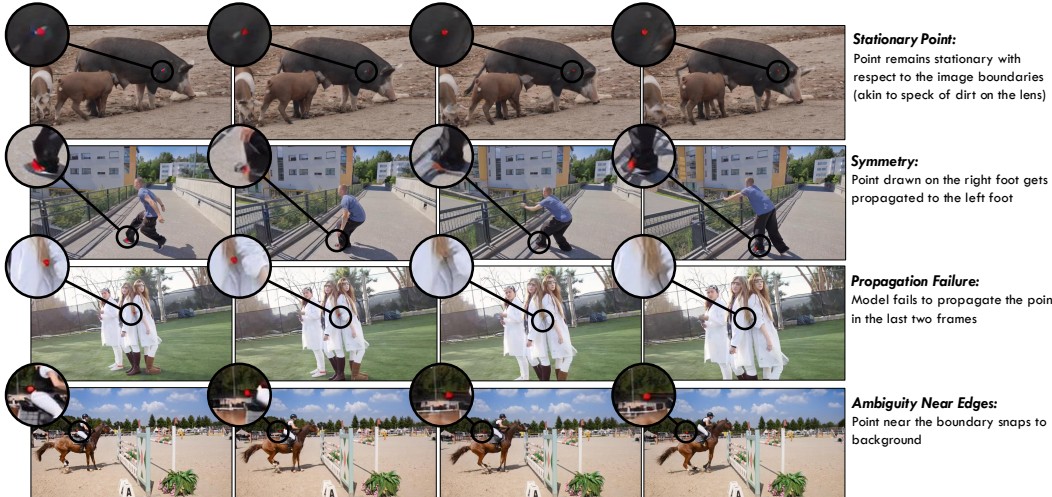

*Stationary Point:*
Point remains stationary with respect to the image boundaries (akin to speck of dirt on the lens)

*Symmetry:*
Point drawn on the right foot gets propagated to the left foot

*Propagation Failure:*
Model fails to propagate the point in the last two frames

*Ambiguity Near Edges:*
Point near the boundary snaps to background

Figure 7: **Generation Failures.** Typical failure cases in video generation: (1) *Stationary Point:* The red dot remains fixed relative to image boundaries, resembling lens dirt. (2) *Symmetry Confusion:* Symmetrical objects (e.g., left and right body parts) cause point propagation errors, likely due to compressed latent representations. (3) *Propagation Failure:* The red dot vanishes across consecutive frames. (4) *Edge Ambiguity:* The red dot, near boundaries, shifts to the background.

## 7 CONCLUSION

We have shown that a video diffusion model, when carefully prompted, can mark the location of a point as it moves through a scene over time. We use this idea to create a simple point tracker, which obtains surprisingly effective tracking results, outperforming previous zero-shot approaches. We see our work as opening two new directions. The first is expanding the number of ways that one can adapt large pretrained video diffusion models to new tasks, such as through prompting schemes that go beyond the use of language. Second, our work shows that video generative models are a useful source of pretraining for tracking. We therefore see our work as a step toward unifying video generation and tracking.

## 8 ACKNOWLEDGEMENTS

We would like to thank Adam Harley, Dan Yamins, and Xuanchen Lu for helpful discussions and feedback on the paper. Daniel Geng was at the University of Michigan when he contributed to the project. This work was supported by the Advanced Research Projects Agency for Health (ARPA-H) under award #1AY2AX000062. This research was funded, in part, by the U.S. Government. The views and conclusions contained in this document are those of the authors and should not be interpreted as representing the official policies, either expressed or implied, of the U.S. Government.

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
