# A    QUANTITATIVE RESULTS ON TAP-VID

Table 6 presents results on TAP-Vid Kubric (using a subset of 30 videos) with our method based on the Wan2.1-14B model. Our approach outperforms zero-shot baselines, consistent with the results reported in Table 1 of the main paper.

However, the overall performance on Kubric is comparatively lower, likely due to the dataset's synthetic nature. The scenes are generated using a graphics simulator and typically consist of simple environments with basic textures and objects exhibiting non-natural, erratic motion, as illustrated in Fig. 8. These characteristics introduce challenges for faithful video re-generation, which in turn impacts the accuracy of point propagation and tracking.

| Method | Supervision | TAP-Vid Kubric | | |
|---|---|---|---|---|
| | | AJ ↑ | $< \delta_{\text{avg}}^{x}$ ↑ | OA ↑ |
| RAFT (Teed & Deng, 2020) | | 68.50 | 83.01 | 89.94 |
| TAP-Net (Doersch et al., 2022) | | 68.22 | 79.87 | 93.35 |
| TAPIR (Doersch et al., 2023) | Supervised | 87.88 | 93.99 | 96.09 |
| CoTracker3 (Karaev et al., 2024b) | | 76.99 | 92.35 | 92.35 |
| TAPNext (Zholus et al., 2025) | | 80.91 | 87.03 | 97.16 |
| GMRW (Shrivastava & Owens, 2024) | Self-Sup. | 55.04 | 72.22 | 84.67 |
| Opt-CWM (Stojanov et al., 2025) | | 60.11 | 77.24 | 85.62 |
| DINOv2+NN (Oquab et al., 2023) | | 20.10 | 40.25 | 53.27 |
| DIFT (Tang et al., 2023) | | 25.93 | 40.12 | 74.08 |
| SD-DINO (Zhang et al., 2023a) | Zero-Shot | 28.89 | 47.11 | 47.10 |
| Ours | | 31.51 | 38.42 | 53.23 |
| Ours (upsampled) | | 33.55 | 40.02 | 54.80 |

Table 6: **TAP-Vid Kubric Results.** We show results on TAP-Vid Kubric with *first* sampling strategy.

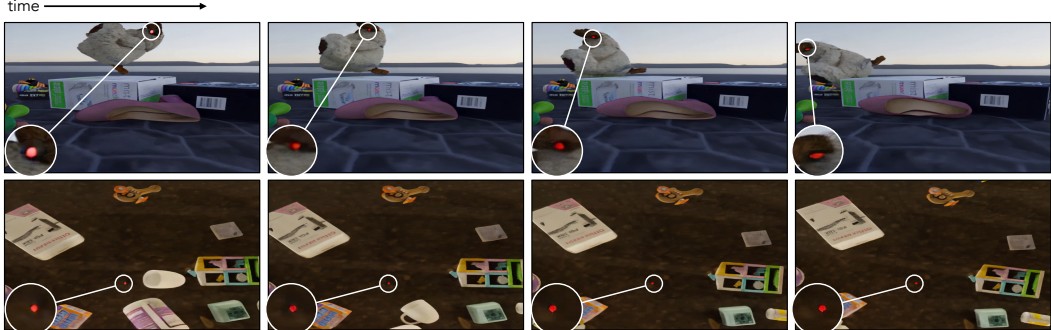

Figure 8: **Qualitative Results on TAP-Vid Kubric.** The top row shows a successful example of point propagation. In contrast, the bottom row illustrates a failure case where the point is not propagated due to the surface having very low texture.

## A.1    ABLATIONS

**Tracker Ablations.** We ablate key components of our tracking pipeline. First, we run the tracker without any enhancements on the generated videos. Adding a local search window around the previously detected point provides a small improvement, especially under occlusion. Gradually expanding the search radius when the query point becomes occluded yields further gains. We then introduce a position–refinement step that averages the coordinates of all red pixels within a fixed neighborhood around the predicted point, achieving the best overall performance. Finally, replacing the HSV color space with LAB causes a slight drop in accuracy, indicating that HSV is better suited for red-dot detection in our setup. Results are shown in Table 7.

| Color space | Local search window | Occlusion based search radius | Average over color pixels | TAP-Vid DAVIS | | |
|---|---|---|---|---|---|---|
| | | | | AJ $\uparrow$ | $< \delta_{\text{avg}}^x \uparrow$ | OA $\uparrow$ |
| HSV | | | | 35.80 | 53.15 | 81.79 |
| HSV | ✓ | | | 38.92 | 53.55 | 84.92 |
| HSV | ✓ | ✓ | | 39.08 | 54.57 | 85.07 |
| HSV | ✓ | ✓ | ✓ | 42.70 | 59.26 | 85.14 |
| LAB | ✓ | ✓ | ✓ | 42.30 | 57.81 | 84.84 |

Table 7: **Tracker Ablations.** (Sec. 3.2). We assess local search window, adaptive radius for occlusions, averaging red pixel positions, and performance across HSV vs. LAB color spaces.

**Additional ablations.** We further assess model hyperparameters on a subset of TAP-Vid DAVIS videos (Table 8). We ablate the parameter $\lambda$ (Eq. 5, main paper), which weights the noise estimate from the edited image. The best performance occurs at $\lambda = 8$. Table 9 reports results when varying the marker color. While our approach is robust to different marker colors, using red provides a slight performance gain.

| Method | TAP-Vid DAVIS | | |
|---|---|---|---|
| | AJ $\uparrow$ | $< \delta_{\text{avg}}^x \uparrow$ | OA $\uparrow$ |
| $\lambda = 4$ | 34.60 | 52.48 | 77.94 |
| $\lambda = 8$ | 35.54 | 52.98 | 78.80 |
| $\lambda = 11$ | 32.82 | 52.08 | 75.66 |
| $\lambda = 14$ | 31.92 | 52.13 | 74.09 |

Table 8: **Counterfactual Enhancement Guidance.** We present ablation results for different values of $\lambda$, which controls the influence of the noise estimate from the edited image (with the colored dot) in counterfactual enhancement guidance.

| Color | TAP-Vid DAVIS | | |
|---|---|---|---|
| | AJ $\uparrow$ | $< \delta_{\text{avg}}^x \uparrow$ | OA $\uparrow$ |
| Red | 48.60 | 63.47 | 85.75 |
| Blue | 46.51 | 60.80 | 84.08 |

Table 9: **Marker color.** We use different marker colors as prompt to show that our approach is invariant to marker color.

### A.2  V-BENCH SCORES

Table 10 shows tracking performance alongside VBench (Huang et al., 2024) scores for Wan2.1 (1.3B and 14B variants), and CogVideoX (Yang et al., 2024b). VBench I2V benchmark evaluates the generation quality of image-conditioned video models. Tracking and generation quality both improve progressively from CogVideoX to Wan2.1-1.3B and further to Wan2.1-14B. We attribute this to the higher video generation quality—reflected in the superior VBench scores—which suggests that better generative models can directly boost tracking accuracy.

| Method | TAP-Vid DAVIS | VBench |
|---|---|---|
| | AJ $\uparrow$ | Total Score |
| CogVideoX1.5-5B (Yang et al., 2024b) | 24.15 | 71.58 |
| Wan2.1-1.3B (Wang et al., 2025) | 44.58 | 83.26 |
| Wan2.1-14B (Wang et al., 2025) | 48.60 | 86.66 |

Table 10: **VBench (Huang et al., 2024) results.** We show VBench numbers for the different video models used.

## B  IMPLEMENTATION DETAILS

### B.1  VIDEO PREPROCESSING

**Color Rebalancing.** Our tracker identifies red pixels in each frame as predicted points. To avoid false positives, we first remove red pixels from the original frame. We convert the frame to the HSV color space and detect pixels whose hue values fall within $[-30°, 10°]$, and whose saturation and value lie inside an ellipse with semi-major and semi-minor axes $r_1 = 80$, $r_2 = 30$, centered at $(255, 255)$. For detected red pixels, we clip the saturation to a maximum of 80, effectively desaturating them.

**Padding Input Video.** Both Wan and CogVideoX require that the input video contains $4T + 1$ frames. To satisfy this constraint, we pad the input by repeating the last frame until this condition is met. After re-generation, the added frames are removed to restore the original length.

**Video Upscaling.** We observe that using high-resolution videos improves point propagation, reducing generation artifacts and minimizing drift. To upscale the input videos, we use Upscale-A-Video (Zhou et al., 2024), a diffusion-based video upscaling method. Starting from $256 \times 256$ input resolution (from TAP-Vid), we upscale to $1024 \times 1024$ using Upscale-A-Video, then downscale to $480 \times 832$ to match the video model's expected resolution. For final tracking evaluation, we resize the output back to $256 \times 256$.

## B.2   POINT PROPAGATION

As described in Sec. 3.1, we use SDEdit with a denoising strength $\gamma = 0.5$ to control the signal-to-noise ratio. The diffusion timestep $t$ is calculated based on $\gamma$ and the total number of diffusion steps $T$:

$$t = \lfloor \gamma \cdot T \rfloor \tag{7}$$

**Counterfactual Enhancement Guidance** To enhance the effect of the guidance from the edited image (with a colored dot), we use Eq. 5 (main paper) to compute the noise estimate. In our experiments, we follow the traditional classifier-free guidance scheme, where the guidance weight $\lambda$ is set to 8.

## B.3   TRACKER

### B.3.1   MARKER DETECTION

To identify the marker in the generated image, we perform color thresholding in the HSV color space. Specifically, we define the hue range as (H-10, H+5), the saturation range as (150, 255), and the value range as (150, 255). A pixel is considered as a potential marker pixel if its HSV components fall within this interval.

### B.3.2   LOCAL SEARCH AND OCCLUSION HANDLING

To effectively locate the marker in each frame, we constrain our search for red pixels to a circular region of radius $r$ centered at the previous detection. By default, this search radius is set to $r_{\text{default}} = 90$. If an occlusion is detected in the previous frame, we expand the search region to accommodate the increased positional uncertainty:

$$r = \min(r_{\text{default}} \times 1.1, r_{\text{max}}) \tag{8}$$

where $r_{\text{max}} = 150$. Once the marker is successfully detected again, we reset $r$ to its default value to maintain efficiency and avoid spurious detections.

### B.3.3   CENTER ESTIMATION

After identifying candidate red pixels, we first select the one closest to the previous detection as an anchor. Around this anchor point, we examine a 20-pixel radius to gather nearby red pixel detections. The final predicted tracking point for the current frame is computed as the average position of these collected pixels. This averaging process produces a stable and consistent estimate for the red blob's center, leading to robust and accurate tracking across frames.