# OpenReview forum: "Point Prompting: Counterfactual Tracking with Video Diffusion Models"
_ICLR.cc/2026/Conference — ICLR 2026 Poster_

### Official Review · Reviewer_nWQb · 2025-10-28

**Soundness:** 3
**Presentation:** 3
**Contribution:** 2
**Rating:** 6
**Confidence:** 3

**Summary:**

This paper proposes Point Prompting, a novel method for zero-shot point tracking using pre-trained I2V diffusion models. It achieves strong performance without any training by adding a visual marker to the first frame and leverage the internal priors of the model to propagate the marker through the generated video.

**Strengths:**

1. Proposed method is novel and simple. It utilizes the existing priors of video diffusion models without requiring costly fine-tuning or specialized architectures.

2. The paper provides thorough ablation studies that validate the contribution of its different components.

3. This paper demonstrates the model-agnostic performance covering diffusion model (CogVideo X) and flow-based model (Wan 2.1 and 2.2)

**Weaknesses:**

1. A major practical limitation is the computational expense. As noted by the authors in L338-339, tracking a single point requires take 7 to 30 minutes. This makes the approach impractical and unfeasible for large-scale offline analysis.

2. It is unclear if the method can handle tracking multiple points simultaneously.

3. More analytical experiments are needed. For instance, the paper would benefit from an attention-based analysis to explore how the model focuses on the point, or an investigation into how tracking paths vary with different random seeds.

4. I wonder how the performance would be affected by using DDIM inversion for the video generation process, rather than the simpler noising approach used.

**Questions:**

1. The explanation of the evaluation metrics (Positional Accuracy, Occlusion Accuracy, and Average Jaccard) could be more detailed.

---

> ### Author Response · Authors · 2025-12-03
>
> ## 1. Computational Cost.
>
> To address efficiency concerns, we first show that our zero-shot method can be distilled into a fast feed-forward tracker. We train a CoTracker-style model using **pseudo-trajectories generated by our Wan2.1-1.3B model**, requiring no human-labeled data:
>
> | Model                          | AJ    | PA    | OA    |
> |--------------------------------|-------|-------|-------|
> | CoTracker (distilled)          | 37.17 | 53.12 | 84.24 |
> | Wan2.1 1.3B model (teacher)    | 38.78 | 54.75 | 85.00 |
>
>
>
> These results indicate that our zero-shot method can be **distilled into an efficient, real-time tracker**, providing a practical path for deployment that avoids diffusion inference entirely. The distilled model offers a natural way to use our approach in scenarios where speed is essential.
>
>
> We also report detailed runtime measurements of the diffusion-based method across model sizes and denoising steps:
>
>
> | # | Model         | Denoising | AJ    | Inference Time |
> |---|---------------|-----------|-------|----------------|
> | 1 | Wan 2.1 14B   | 50 steps  | 48.60 | 30 mins        |
> | 2 | Wan 2.1 14B   | 25 steps  | 49.97 | 9.5 mins       |
> | 3 | Wan 2.1 1.3B  | 10 steps  | 43.86 | 76 secs        |
>
>
>
> Our best model (Row 2) takes **9.5 minutes per point**, and our fastest model (Row 3) takes **76 seconds per point**. These numbers reflect the raw diffusion inference cost; however, the diffusion model is not intended to serve as the deployed tracker. It can instead be used to **extract supervisory signal**, while the distilled feed-forward model is the one that would be used in practice.
>
>
>
> ## 2. Tracking multiple points.
>
> While our current implementation processes one query point per inference pass, the framework naturally extends to multi-point tracking by encoding points with distinct, high-contrast colors and updating the tracker to distinguish between hue ranges.
>
> However, we *intentionally did not introduce additional engineering* for multi-point tracking. Our goal in this work is to isolate and study the *emergent tracking behavior* of video diffusion models, not to build a full tracking system. Adding a more sophisticated multi-point mechanism (for example, one based on SuperPoint-style descriptors [1]) would introduce confounding factors and make it unclear whether improvements come from the generative model or from the added tracking machinery. Thus, our architecture-agnostic approach serves as a focused proof-of-concept for the latent tracking capabilities of video generators. We leave the engineering optimizations to future work.
>
>
> ## 3. Different random seeds.
>
> We tried a small-scale experiment using different random seeds and observed no meaningful change in the resulting tracking trajectories. Both the coarse propagation and the refinement stages remained stable across seeds, indicating that the method is robust to stochastic variation in the diffusion sampling process.
>
> ### Attention-based analysis.
>
> While we appreciate the suggestion, our method is designed to be fully architecture-agnostic and operates entirely in pixel space without accessing or modifying internal components such as attention maps. Introducing attention-based analysis would run counter to the purpose of the work, which is to show that tracking emerges without requiring model introspection or architectural assumptions. Our goal was to demonstrate a clean, model-agnostic phenomenon rather than to analyze a specific architecture's internals.
>
>
>
> ## 4. DDIM inversion in the video generation process.
> We thank the reviewer for the suggestion. We could try using that in future work.
>
>
>
> ## 5. Evaluation metrics details.
> Thanks, we will expand the descriptions of metrics in a revision.
>
>
> ## References:
>
> [1] Daniel DeTone, Tomasz Malisiewicz, and Andrew Rabinovich. "Superpoint: Self-supervised interest point detection and description." Proceedings of the IEEE conference on computer vision and pattern recognition workshops. 2018.

---

### Official Review · Reviewer_sTu5 · 2025-10-28

**Soundness:** 4
**Presentation:** 4
**Contribution:** 2
**Rating:** 6
**Confidence:** 4

**Summary:**

The paper explores repurposing large-scale pre-trained video generative models for object tracking. It leverages the rich spatio-temporal representations learned by these models and adapts them in a training-free manner to track targets across frames. The paper is well-written, logically structured, and presents clear quantitative results that complement the experimental tables.

**Strengths:**

* Clearly motivated and interesting idea of repurposing large-scale pre-trained video generative models for object tracking.
* Intuitive and easy-to-understand method.
* Comprehensive experiments with both quantitative metrics and qualitative results demonstrating the approach’s effectiveness.

**Weaknesses:**

* Runtime comparison of each component in the pipeline (i.e., the time cost of steps listed in Table 4)?
* What are the effects of different denoising steps? Was a simple Euler solver used, and what is the time schedule?
* Is the tracking performance evaluated on real-world data or synthetic data (e.g., self-generated clips)?
* Can you confirm whether the models are run in image-to-video mode?
* Can you compare with the latest works on flow-matching model editing? [1-2]


Ref:\
[1] Jiao, G., Huang, B., Wang, K.C. and Liao, R., 2025. Uniedit-flow: Unleashing inversion and editing in the era of flow models. arXiv preprint arXiv:2504.13109.\
[2] Kulikov, V., Kleiner, M., Huberman-Spiegelglas, I. and Michaeli, T., 2025. Flowedit: Inversion-free text-based editing using pre-trained flow models. In Proceedings of the IEEE/CVF International Conference on Computer Vision (pp. 19721-19730).

**Questions:**

Please see the [Weakness] section.

---

> ### Author Response · Authors · 2025-12-03
>
> ## 1. Runtime comparison of each component in the pipeline
>
>
> | Component                   | Time (approx.) | Percentage |
> |-----------------------------|----------------|------------|
> | Generate video (coarse)     | 1256 s         | 49.92%     |
> | Generate video (refinement) | 1256 s         | 49.92%     |
> | Extract tracks              | 3 s            | 0.12%      |
> | Color rebalancing           | 1 s            | 0.04%      |
> | **Total**                   | ~2516 s        | **100%**   |
>
>
> As shown above, nearly all computation time is dominated by the two video-generation passes, while the remaining components contribute negligibly to the runtime.
>
>
> ## 2. Effect of denoising steps and solver details
>
> The effect of different denoising steps is shown below:
>
> | Model       | Steps    | AJ      | PA      | OA      | Inference time |
> |-------------|----------|---------|---------|---------|----------------|
> | Wan2.1 1.3B | 10 steps | 0.4386  | 0.5806  | 0.8444  | 76 seconds     |
> | Wan2.1 1.3B | 25 steps | 0.4445  | 0.5876  | 0.8535  | 191 seconds    |
> | Wan2.1 14B  | 10 steps | 0.4708  | 0.6204  | 0.8451  | 229 seconds    |
> | Wan2.1 14B  | 25 steps | 0.4997  | 0.6460  | 0.8577  | 575 seconds    |
>
>
> Increasing the number of denoising steps yields modest improvements in AJ, PA, and OA, at the cost of increased runtime.
>
> ### Solver and time schedule.
>
>  We use the same solver and timestep schedule as the underlying video diffusion model, without modification. Wan2.1 uses a flow-matching based solver (FlowDPMSolverMultistepScheduler). During training, the model follows the rectified-flow interpolation schedule:
>
> $$x_t = t \cdot x_1 + (1 - t) \cdot x_0,$$
>
> where $x_0​$ is noise and $x_1$​ is the clean signal. We simply reuse this native schedule during inference.
>
> ## 3. Evaluation on real-world vs. synthetic data
>
> We evaluate our method on both real-world and synthetic datasets.
>
> - Real-world: TAP-Vid Kinetics and TAP-Vid DAVIS (reported in Table 1 of the main paper).
> - Synthetic: TAP-Vid Kubric (reported in Table 5 in Appendix C).
>
> ## 4. Image-to-Video mode
>
> Yes, we operate in the Image-to-Video (I2V) mode. Our method conditions generation on the edited first frame that contains the query point. We additionally use the unedited original first frame as a negative prompt, which provides counterfactual guidance and prevents the model from removing the marker during generation. This setup ensures that the marker is consistently propagated through time.
>
>
> ## 5. Comparison with recent flow-matching editing methods
>
> While UniEdit-Flow [1] and FlowEdit [2] are strong advances in flow-based **editing**, they are fundamentally different from our setting along two key axes:
>
> ### Modality (text-driven vs. image-driven).
>
>  UniEdit-Flow and FlowEdit are designed for *text-based semantic editing,* where the goal is to modify video appearance according to a prompt (e.g., changing style or attributes). Our method is *image-driven*: we directly modify the pixel space of the first frame by adding a marker, and we prompt the model to propagate that marker with accurate spatial propagation across the video.
>
> ### Task objective (editing vs. tracking).
>
> Flow-based editing methods focus on altering the video while keeping its structure consistent. Our task is the inverse: we aim to preserve the entire video while inserting a marker whose motion we want to recover as a trajectory. This requires fine-grained correspondence, not semantic transformation.
>
> Because of these differences in modality and objective, text-based inversion and editing frameworks such as UniEdit-Flow and FlowEdit do not apply to our fine-grained point tracking task.
>
>
>
> ## References:
>
> [1] Jiao, G., Huang, B., Wang, K.C. and Liao, R., 2025. Uniedit-flow: Unleashing inversion and editing in the era of flow models. arXiv preprint arXiv:2504.13109.
>
> [2] Kulikov, V., Kleiner, M., Huberman-Spiegelglas, I. and Michaeli, T., 2025. Flowedit: Inversion-free text-based editing using pre-trained flow models. In Proceedings of the IEEE/CVF International Conference on Computer Vision (pp. 19721-19730).

---

### Official Review · Reviewer_duWk · 2025-11-01

**Soundness:** 1
**Presentation:** 2
**Contribution:** 1
**Rating:** 2
**Confidence:** 5

**Summary:**

This paper investigates whether pretrained image-conditioned video diffusion models exhibit emergent point-tracking capabilities. The authors propose a “point prompting” technique in which a red dot is placed on the first frame of a real video, the video is regenerated using SDEdit, and the propagated dot is tracked via color-based detection. Several heuristics (color rebalancing, negative prompting, and inpainting refinement) are introduced to stabilize the dot’s visibility across frames. Experiments on TAP-Vid show improvements over image-based zero-shot tracking baselines, and the authors claim that these results indicate temporal reasoning and object permanence in video diffusion models.

**Strengths:**

- Limitations are clearly articulated, with helpful visual examples that make failure modes easy to interpret.
- Unlike some zero-shot correspondence approaches, the work attempts to handle occlusion.

**Weaknesses:**

### 1. Limited novelty and lack of conceptual advancement

The core claim that video diffusion models contain emergent temporal correspondences, has already been demonstrated by prior work such as DiffTrack [1]. The statements in lines 35–36 and 92–93 suggest novelty in analyzing emergent tracking in video diffusion models, but the conceptual contributions closely follow existing findings and provide little new insight into the temporal behavior of DiT-based models.


### 2. The method is not suitable for either analysis or tracking

(1) Misalignment with analysis goals

The methodology relies heavily on pixel-space operations rather than model-level signals. The pipeline includes removing all red pixels from the input, adjusting global color balance, reducing marker saturation after generation, performing inpainting refinement when the dot drifts. These operations substantially alter the video content and disconnect the analysis from the model’s inherent behavior. Because the method depends on multiple rounds of video regeneration, it does not capture the model’s natural temporal consistency. Table 4 further shows that performance sharply degrades without the heuristic refinements, indicating that consistency comes from the heuristics rather than from the generative model itself.

(2) Inefficiency as a point tracking method

The method tracks the dot purely based on pixel color, ignoring positional encoding and geometry, and failing on rapid motion. Because each point requires re-generating the entire video often more than once, the pipeline is extremely inefficient.

Overall, the method behaves more like a handcrafted video-editing pipeline than a principled analysis of temporal correspondences.

### 3. Writing and presentation issues
- The definition of the tracking problem is unclear (pixel-level vs. semantic vs. object-level).
- Lines 36–38 contain vague phrasing (“high-level understanding tasks”) and ambiguous pronoun ("these capabilities") use.
- The Related Work section blends supervised, self-supervised, and counterfactual modeling approaches without clear structure.

### 4. Missing comparisons and limited generalization
- The most relevant baseline, DiffTrack [1], is not included in quantitative comparisons, which weakens the empirical evaluation.
- The method applies only to image-conditioned video diffusion models and cannot be used for text-to-video models, contradicting claims of architectural generality.
- Experiments do not analyze how point radius interacts with input resolution, potentially biasing comparisons across models.


[1] Nam, Jisu, et al. "Emergent Temporal Correspondences from Video Diffusion Transformers." (NeurIPS 2025)

**Questions:**

- Lines 107–108 claim that DINOv2 has been adapted for temporal correspondence. What specific prior work supports this claim? A citation is required.
- The paper reports (line 338-341) runtime for a single 50-frame generation, but the full pipeline requires at least two rounds (generation + refinement). What is the actual cost per tracked point?
- In Table 3, what exactly is the configuration represented by the second row (“DAVIS 256×256 up.”)?

---

> ### Author Response · Authors · 2025-12-03
>
> ## 1. Novelty and comparison to DiffTrack [1]:
>
> We are surprised that we are being asked to compare against DiffTrack, as it clearly falls under *concurrent work* according to the ICLR reviewer policy. The policy states:
>
>
>
>  > “We consider papers contemporaneous if they are published within the last two months. That means, since our full paper deadline is September 24, if a paper was published (i.e., at a peer-reviewed venue) on or after July 24, 2025, authors are not required to compare their own work to that paper. Note that arXiv is not considered a peer-reviewed venue.”
>
>
> DiffTrack was posted on arXiv on **June 20, 2025** and was accepted to NeurIPS on **September 18, 2025**, which places its publication well within the “concurrent work” window defined by the ICLR policy. Under this policy, we are not required to compare against it.
>
> That said, our approach is conceptually and practically different from DiffTrack. DiffTrack relies on a **highly architecture-specific analysis**, identifying particular DiT layers and heads that must be guided via cross-attention. Their method depends on **CoTracker supervision** to determine which components to manipulate, and the resulting technique is tightly tied to the internal structure of a specific video diffusion backbone. In contrast, our method operates **entirely in pixel space**, without inspecting or modifying any internal model components. This makes our approach broadly applicable to any image-conditioned video diffusion model, without architecture-specific assumptions.
>
> Furthermore, DiffTrack does not explicitly handle occlusions, whereas we evaluate and quantify occlusion accuracy (OA) and show that our method is robust to occlusion and disocclusion events. On TAPVid-DAVIS, we outperform DiffTrack with an **AJ of 48.78 vs. 46.9**, and importantly, our tracked points remain consistent even through occlusion boundaries.
>
>
> ## 2.1 Alignment with analysis goal:
>
> We are puzzled by this remark. The reviewer’s concern appears to stem from a misunderstanding of our design goal. Our goal is to develop a **general-purpose method for prompting a pretrained video diffusion model to track a marker**, *without modifying or accessing internal model features*. All tracking behavior emerges from the model’s own generative dynamics. This is aligned with prior work on visual prompting [5, 10, 11, 12] where pixel-level modifications are standard because they directly interface with the model’s input.
>
> ### Operating in pixel space is intentional and preserves model behavior.
>  We purposely do not alter internal features or modify the model’s architecture. Instead, we rely on **counterfactual enhancement guidance**, which encourages the model to propagate the marker through time. Prompting the model in pixel space allows us to analyze how the pretrained video model itself maintains temporal consistency. In fact, we argue that the marker reliably propagates even without feature-level interventions *supports* the claim that video diffusion models possess strong inherent temporal coherence.
>
> ### Video content is minimally altered.
>  The small color-balancing step affects **only the marker color**, ensuring it does not clash with background pixels for simple color-based extraction. It does *not* change the underlying video content. After the refinement stage, the regenerated video remains visually identical to the input video everywhere except for the marker itself, and a very small region around it used for local refinement. Thus, the analysis remains tightly coupled to the model’s behavior; not disconnected from it.
>
> ### The method is not heuristic-dependent.
>  Our two-stage process, coarse propagation followed by local refinement, is a standard **coarse-to-fine strategy**, widely used in tracking methods [2, 3, 4]. The reviewer states that performance “sharply degrades” without heuristics, but this is inaccurate:
>
> - Removing the refinement step lowers AJ by only **5.9 points**, which is far from a collapse.
>
>
> - Table 4 also shows that **counterfactual enhancement** is a key component of the method and is not a minor heuristic. It meaningfully improves propagation quality (by 20.67 points) and directly reflects the generative model’s response to counterfactual prompts.

---

> ### Author Response · Authors · 2025-12-03
>
> ## 2.2 Efficiency of point tracking:
>
> The reviewer’s claim that our method “ignores positional encoding and geometry” is inaccurate. These aspects are handled directly by the **pretrained video model**. Wan2.1, which we use as the backbone, applies **RoPE positional encoding** to latent tokens before processing them through the DiT transformer. This allows the model to capture positional structure, geometry, and scene dynamics. Our method leverages these built-in capabilities rather than bypassing them.
>
> To address efficiency, we first show that our zero-shot method can be distilled into a fast feed-forward tracker. Following the same strategy as OPT-CWM, we train a CoTracker-style model using **pseudo-trajectories generated by our Wan2.1-1.3B model**, requiring no human-labeled data. The distilled model achieves:
>
> | Model                          | AJ    | PA    | OA    |
> |--------------------------------|-------|-------|-------|
> | CoTracker (distilled)          | 37.17 | 53.12 | 84.24 |
> | Wan2.1 1.3B model (teacher)    | 38.78 | 54.75 | 85.00 |
>
> This shows that our zero-shot method can be converted into a **fast, lightweight point-tracking model** with performance close to the teacher, while completely avoiding the cost of diffusion inference at test time.
>
> Regarding computational efficiency of the diffusion-based approach, we provide concrete measurements across model sizes and denoising steps:
>
>
> | # | Model         | Denoising | AJ    | Inference Time |
> |---|---------------|-----------|-------|----------------|
> | 1 | Wan 2.1 14B   | 50 steps  | 48.60 | 30 mins        |
> | 2 | Wan 2.1 14B   | 25 steps  | 49.97 | 9.5 mins       |
> | 3 | Wan 2.1 1.3B  | 10 steps  | 43.86 | 76 secs        |
>
>
> Our best configuration (Row 2) takes **9.5 minutes per point**, and our fastest setting (Row 3) takes **76 seconds per point**. These runtimes are consistent with Opt-CWM [5], a next-frame prediction method that performs point propagation at similar computational cost (reported runtime: **50 minutes for a 50-frame video**). Opt-CWM generates **one frame at a time**, whereas our method generates the **entire video in a single pass**. Both approaches require running a large video model per point in a zero-shot setting, so runtimes of this scale are expected.
>
>
> ## 3.1 Clarification about the definition of tracking
>
> We are surprised that the reviewer found the definition of the tracking problem unclear. *Point tracking* is now a widely studied and well-established research area [2, 3, 4, 5, 6, 7]. Our paper explicitly states that we focus on **point tracking**, not semantic or object-level tracking. This is mentioned repeatedly throughout the paper, including the abstract [L015] and in multiple other sections [L073, L084, L180, L227, L247]. Given these explicit mentions, the intended formulation should already be reasonably clear.
>
>
> ## 3.2 Clarification about “high-level understanding tasks” and the related work section.
> Thank you for the suggestion, we will revise the phrasing in the next draft and also restructure the related work section.
>
>
> ## 4.1 Comparison to DiffTrack [1].
> Already answered in Point 1 as concurrent work.
>
> ## 4.2 Using text-to-video models
>
> Our method is **not inherently limited** to image-conditioned video diffusion models. While the counterfactual enhancement step specifically relies on modifying the conditional image, the underlying idea of propagating a pixel-space marker through the video does not depend on image conditioning. In principle, the same mechanism can be applied to text-to-video models by inserting the marker into the **first generated frame**, adding noise, and then performing an SDEdit-style denoising process. This would prompt the text-to-video model to carry the marker forward in time, analogous to how our method operates in the image-conditioned setting. This does not contradict our claim of architectural generality; rather, it clarifies that the counterfactual enhancement technique is specific to image-conditioned models, while the core idea can extend more broadly.

---

> ### Author Response · Authors · 2025-12-03
>
> ## 4.3 Effect of resolution and query point radius on performance.
>
> We believe the reviewer’s concern about potential bias from point radius versus input resolution is addressed directly in our experiments. As shown in **Table 3**, higher-resolution source videos do lead to better performance, but **all videos are ultimately resized to the model’s native resolution (480×832 for Wan)** before any processing occurs. This ensures that comparisons across models and inputs are made at a consistent spatial scale.
>
> The effect of the query point radius is analyzed in **Fig. 4 (bottom)**. Importantly, the red marker is always added **after** resizing to 480×832, meaning its size is defined in the model’s coordinate system rather than the original video resolution. Therefore, there is no opportunity for bias arising from interactions between the query radius and the input resolution, the radius is fixed relative to the model input, not the original video.
>
> ## Questions:
> 1. **DINOv2 temporal correspondence.**
> DINO-Tracker [8] had a baseline which adapted DINOv2 for temporal correspondence, we will add a citation.
>
> 2. A full point track requires two passes (generation + refinement). The end-to-end runtime is ~60 min per point with Wan 2.1-14B (50 steps) for the best model and ~2 min per point with Wan 2.1-1.3B (10 steps) for the fastest model.
>
> 3. **“DAVIS 256×256 up.”** refers to the following configuration:
> We take the TAPVid-DAVIS videos, which are originally 256×256, and upsample them to 480×832 using the video upsampling method Upsample-A-Video [9]. The upsampled videos are then used as input to the video diffusion model, which natively operates at 480×832 resolution.
>
>
>
>
> ## References:
>
> [1] Emergent Temporal Correspondences from Video Diffusion Transformers, NeurIPS 2025.
>
> [2] TAPIR: Tracking any point with per-frame initialization and temporal refinement, Carl Doersch et al., ICCV 2023.
>
> [3] CoTracker: It is Better to Track Together, Nikita Karaev et al., ECCV 2024.
>
> [4] TAPTR: Tracking Any Point with Transformers as Detection, Hongyang Li et al, ECCV 2024.
>
> [5] Self-Supervised Learning of Motion Concepts by Optimizing Counterfactuals, NeurIPS 2025.
>
> [6] CoTracker: It is Better to Track Together, ECCV 2024.
>
> [7] TAP-Vid: A Benchmark for Tracking Any Point in a Video, NeurIPS 2022.
>
> [8] DINO-Tracker: Taming DINO for Self-Supervised Point Tracking in a Single Video, ECCV 2024.
>
> [9] Upscale-a-video: Temporal-consistent diffusion model for real-world video super-resolution, CVPR 2024.
>
> [10] What does CLIP know about a red circle? Visual prompt engineering for VLMs, ICCV 2023.
>
> [11] Fine-Grained Visual Prompting, NeurIPS 2023.
>
> [12] DiffusionLight: Light Probes for Free by Painting a Chrome Ball, CVPR 2024.

---

### Official Review · Reviewer_yFzS · 2025-11-01

**Soundness:** 3
**Presentation:** 3
**Contribution:** 2
**Rating:** 4
**Confidence:** 4

**Summary:**

This paper proposes a training-free method that adopts video diffusion models for point tracking.  It first puts a colored marker on the query point in the first frame, then asks the video diffusion models to propagate the marker across frames by generating new videos. To avoid the loss of the marker in video generation, this paper proposes to use the unedited video’s initial frame as a negative prompt. The proposed method is evaluated on the TAP-Vid benchmark.

**Strengths:**

1. The idea of using a colored marker to indicate the query point is interesting and insightful.
2. It is also interesting to use an unedited video’s initial frame as a negative prompt to make the marker visible.
3. The paper is well-written and easy to follow.

**Weaknesses:**

1. This paper requires video generation to get the tracking results. What is the computational cost of this method for tracking one point, compared to methods that do not use diffusion models?

2. This approach requires generating a video for each tracked point, which is difficult to use in real applications.

3. Previous work[1] already shows that video diffusion models have an inherent ability for point tracking. Simlilar observation is also proposed in [2].

[1] Emergent Temporal Correspondences from Video Diffusion Transformers
[2] Track4Gen: Teaching Video Diffusion Models to Track Points Improves Video Generation.

4. The tracking performance is much lower than non-diffusion methods such as CoTracker3. Although it performs better than DIFT and SD-DINO, these two methods are not designed for point tracking in videos.  Considering the high computation cost, the tracking accuracy is not good enough.

5.  According to line 305, the proposed method allows the video diffusion models to
>  generate only regions near the potential tracked point.

What if there are occlusions or significant object motions in the videos？

6. The accuracy of the tracking method is limited by the generation ability of video diffusion models, which limits the effectiveness of the proposed method in diverse scenarios. This also raises my concern about the robustness of the proposed method in cases such as
small objects,  corrupted videos, or videos with poor weather, and so on.

7. How does the method track points in long videos that exceed the maximum length supported by video diffusion models?

8. This paper lacks the text “Under review as a conference paper at ICLR 2026” in the header, which is unexpected according to the official template.

**Questions:**

Please refer to the weakness part.

---

> ### Author Response · Authors · 2025-12-03
>
> ## 1. Computational Cost:
>
>
> To address efficiency concerns, we first show that our zero-shot method can be converted into a **lightweight feed-forward tracker**. Following the same strategy as Opt-CWM, we distill our zero-shot tracker into a CoTracker-style model using **pseudo-trajectories generated by our Wan2.1-1.3B model** on 1000 Kinetics videos, requiring no labeled training data. The distilled model achieves:
>
> | Model                          | AJ    | PA    | OA    |
> |--------------------------------|-------|-------|-------|
> | CoTracker (distilled)          | 37.17 | 53.12 | 84.24 |
> | Wan2.1 1.3B model (teacher)    | 38.78 | 54.75 | 85.00 |
>
> This shows that our zero-shot method can be transformed into a **fast, feed-forward point tracker** with performance close to the teacher model, while completely eliminating the cost of diffusion inference at test time.
>
> We also provide a breakdown of the computational cost of the diffusion-based method across model sizes and denoising steps:
>
> | # | Model         | Denoising | AJ    | Inference Time |
> |---|---------------|-----------|-------|----------------|
> | 1 | Wan 2.1 14B   | 50 steps  | 48.60 | 30 mins        |
> | 2 | Wan 2.1 14B   | 25 steps  | 49.97 | 9.5 mins       |
> | 3 | Wan 2.1 1.3B  | 10 steps  | 43.86 | 76 secs        |
>
> Our best-performing configuration (Row 2) takes **9.5 minutes per point**, while our fastest configuration (Row 3) takes **76 seconds per point**. These runtimes are consistent with Opt-CWM [3], a next-frame prediction approach that also propagates points by repeatedly running a large video model (reported runtime: **50 minutes for a 50-frame video**). Opt-CWM generates **one frame at a time**, whereas our method produces the **entire video in a single pass**. Given that both approaches operate in a zero-shot setting and require running the generative model per point, this computational cost is expected.
>
>
> ## 2. Comparison to CoTracker3 and Applicability to Zero-shot Tracking:
> We are surprised that the reviewer directly compares our method’s zero-shot, training-free performance with CoTracker3, a fully supervised model. CoTracker3 is trained on a **massive supervised synthetic dataset**, the Kubric-based dataset containing 6,000 high-resolution sequences specifically rendered for trajectory supervision. Our method, in contrast, requires no training data and operates entirely zero-shot. For this reason, a direct comparison between the two is not a fair evaluation of our approach.
>
> Zero-shot correspondence and zero-shot tracking constitute a well-established research area [3, 5, 10, 11, 12, 13]. Recent work such as **Opt-CWM** [3] follows the same paradigm as ours: using a **next-frame predictor** to propagate points in a zero-shot manner and performing a task similar to ours. Our method’s backbone is a **video generative model**, and as such video models continue to improve, the performance ceiling of our approach will naturally increase as well.
>
> To further address concerns about practical applicability, we show that our zero-shot tracker can be *distilled* into a **fast feed-forward CoTracker-style model using only pseudo-labels generated by our method**. As shown in Rebuttal Point 1, the distilled tracker achieves performance close to its teacher while offering significantly faster inference. This confirms that our method can serve as a strong zero-shot supervisory signal, similar in spirit to Opt-CWM.

---

> ### Author Response · Authors · 2025-12-03
>
> ## 3. Applications:
>
> This concern seems to miss the primary purpose of our method. Our goal here is to develop a *general-purpose way to prompt a pretrained video diffusion model to track a marker*. The method is **not** intended to run diffusion sampling at test time for every point. Rather, its practical value lies in using our zero-shot method to *distill the knowledge of a large pretrained video model into a small, fast feed-forward tracker*. As shown in the distillation results above, the resulting CoTracker-style model achieves performance close to the teacher model, with efficiency comparable to supervised trackers, while requiring no labeled trajectories. Our approach shows that it is possible to extract useful tracking knowledge from large pretrained video models, and that this knowledge can be leveraged to train fast inference models without human supervision.
>
> We are also surprised that the reviewer is evaluating our work strictly as a tracking method. Our paper is not a tracking paper in the traditional sense. It follows the tradition of empirical works such as [5, 6, 7, 8, 9] that investigate surprising emergent capabilities of generative models. Today’s strongest tracking systems rely heavily on supervised trajectory datasets, which are extremely limited and expensive to scale. Progress on tracking benchmarks has slowed in part because of this bottleneck.
>
> In contrast, video generative models continue to scale rapidly, with stronger models appearing every month. It would be a missed opportunity for the community to ignore the possibility that such models could eventually replace or augment human supervision in training trackers. Our result that a video diffusion model can be prompted to perform zero-shot point tracking significantly better than previous zero-shot methods is an important step toward that future. Expecting this empirical finding to immediately translate into a production-ready tracker misunderstands the purpose of the work and how progress in generative-model-driven analysis typically unfolds.

---

> ### Author Response · Authors · 2025-12-03
>
> ## 4.1 Comparison with DiffTrack [1]:
>
> We are surprised that we are being asked to compare to DiffTrack, since it clearly qualifies as concurrent work under the ICLR reviewer policy. The policy states:
>
> > "We consider papers contemporaneous if they are published within the last two months. That means, since our full paper deadline is September 24, if a paper was published (i.e., at a peer-reviewed venue) on or after July 24, 2025, authors are not required to compare their own work to that paper. Note that arXiv is not considered a peer-reviewed venue.”
>
>
> DiffTrack was posted on arXiv on **June 20, 2025** and accepted to NeurIPS on **September 18, 2025**, placing it firmly within the “concurrent work” window. Under ICLR policy, we are *not required* to compare against it.
>
> That said, DiffTrack is fundamentally different from our work. DiffTrack depends on a **highly architecture-specific analysis**, requiring identification of specific DiT layers and heads suitable for cross-attention guidance. Their method relies on **CoTracker supervision** to discover these components and cannot be broadly applied across video diffusion architectures. In contrast, our approach operates **entirely in pixel space**, without accessing or modifying internal model components, making it *directly applicable to a wide range of image-conditioned video diffusion models*.
>
> Furthermore, DiffTrack does not handle occlusions, whereas we explicitly **evaluate** occlusion accuracy (OA). On TAPVid-DAVIS, we even **outperform** DiffTrack with **AJ 48.78 vs. 46.9**, and our method maintains consistent point propagation through occlusions and disocclusions.
>
> ## 4.2 Comparison with Track4Gen [2]
>
> We are puzzled by the suggestion to compare with Track4Gen, since it addresses an entirely different problem. Although both works involve video generation and tracking, the underlying paradigms are **inverse** to one another.
>
> ### Direction of supervision.
>  Track4Gen uses a **tracker to improve a video generator**, whereas our work performs the opposite: we **prompt a pretrained video diffusion model to perform tracking**. Our method is **zero-shot**, requiring no training, supervision, or model modification. Track4Gen, on the other hand, explicitly **trains** a video diffusion model with track-level supervision and uses an **off-the-shelf supervised tracker** during training, making it closer to a supervised point tracking method.
>
> ### Architectural generality.
>  Our approach is **architecture-agnostic** and works with various pretrained models (e.g., Wan2.1, CogVideoX) without internal access. Track4Gen is tightly coupled to specific architectures and, as shown in the paper, is evaluated only on a single SVD-based model [4], requiring nontrivial engineering and retraining for adaptation.
>
> ### Core conceptual difference.
>  Our central claim is that **tracking emerges naturally** as a byproduct of learning to generate high-quality videos. Track4Gen instead investigates the **inverse claim**, that adding specialized tracking supervision can improve video generation. These are orthogonal research questions.
>
>
> ## 5. Clarification about refinement.
>
> The reviewer seems to have misunderstood our refinement step. Our method consists of two stages:
>
> - **Coarse propagation**:
>  In the first step, the video model re-generates the entire video while conditioning on the initial marker. During this process, the video model naturally **propagates the marker through the video**, producing a coarse trajectory. This coarse trajectory is not restricted to any spatial region; it reflects how the pretrained video model interprets global scene dynamics, including large motions and temporal consistency.
>
> - **Refinement via regional inpainting**:
>  In the second step, we refine the point locations by **inpainting only a small region around the coarse marker location**. This refinement is local and purely spatial; it does not determine whether the point appears in a frame. Instead, it improves localization accuracy based on the spatial context provided by the surrounding video content.
>
> ### Handling of significant motion.
>  If there is large or fast object motion, it is already captured in the **first step**, where the video model propagates the marker throughout the full video. The refinement step does not limit or affect large motions because it operates only around the coarse point that has already been globally propagated. We show this in **Figure 5 (middle row)**.
> ### Handling of occlusions.
>  If the point becomes occluded, the video model will naturally **fail to render the marker** during the first step. Such frames are interpreted as occlusions, and in these cases the refinement step simply does nothing, as there is no coarse location to refine. We show an example of this in **Figure 5 (top row)**.

---

> ### Author Response · Authors · 2025-12-03
>
> ## 6. Performance of tracking using video models and small objects.
>
> The reviewer is concerned that our method’s accuracy depends on the generation ability of the underlying video diffusion model. We view this dependency as a **strength**, not a weakness. Video generative models are improving at an extremely rapid pace due to larger and more diverse training corpora, better architectures, and scaled training regimes. As these models continue to advance, the performance of our method will **naturally scale with them**, without requiring any additional supervision or engineering.
>
>
> In contrast, today’s strongest tracking methods rely heavily on **supervised trajectory datasets**, which are scarce, expensive to produce, and fundamentally difficult to scale. Progress on many tracking benchmarks has slowed precisely because of this bottleneck. Meanwhile, video generative models have a clear and accelerating scaling path, with stronger models appearing every month. Ignoring the possibility that such models will eventually be used as supervision sources for tracking would be shortsighted. Our finding that pretrained video diffusion models can be prompted to perform point tracking **far better than previous zero-shot methods** is an important step in this direction. Expecting this empirical observation to immediately yield a production-ready tracker is unrealistic and inconsistent with how foundational progress typically emerges in generative modeling research.
>
>
> Regarding robustness to small objects: our method tracks a **single marker**, not the full object, which makes it more resilient in cases where the object is small, thin, or partially occluded. For example, in **Figure 5 (middle row)**, we place the marker on the narrow bicycle stem where it connects to the wheel, and the model successfully tracks it even through occlusions. This shows that our approach is not limited to large or prominently featured objects.
>
> ## Minor:
>
> 7. Both **Wan2.1** and **CogVideoX** are DiT-based video diffusion models that support adaptive video lengths. They do not impose a fixed temporal window, so our method can track points in long videos simply by feeding the full sequence to the model.
>
> 8. Missing “Under review as a conference paper at ICLR 2026” – Thank you for pointing this out. We will correct the header in the next revision.
>
> ## References:
>
> [1] Emergent Temporal Correspondences from Video Diffusion Transformers, NeurIPS 2025.
>
> [2] Track4Gen: Teaching Video Diffusion Models to Track Points Improves Video Generation, CVPR 2025.
>
> [3] Self-Supervised Learning of Motion Concepts by Optimizing Counterfactuals, NeurIPS 2025.
>
> [4] Stable Video Diffusion: Scaling Latent Video Diffusion Models to Large Datasets, Andreas Blattmann et al, arXiv 2023.
>
> [5] DIFT, NeurIPS’23.
>
> [6] DiffusionLight: Light Probes for Free by Painting a Chrome Ball, CVPR’24.
>
> [7] Contrastive Region Guidance: Improving Grounding in Vision-Language Models without Training, ECCV’24.
>
> [8] Fine-Grained Visual Prompting, NeurIPS’23.
>
> [9] What does CLIP know about a red circle? Visual prompt engineering for VLMs, ICCV’23.
>
> [10] Self-Supervised Any-Point Tracking by Contrastive Random Walks, ECCV 2024.
>
> [11] DINOv2: Learning Robust Visual Features without Supervision, TMLR 2024.
>
> [12] A Tale of Two Features: Stable Diffusion Complements DINO for Zero-Shot Semantic Correspondence, NeurIPS 2023.
>
> [13] Diffusion Hyperfeatures: Searching Through Time and Space for Semantic Correspondence, NeurIPS 2023.

---

### Meta-Review · Area_Chair_cKpx · 2025-12-13

**Summary:**

Strengths:
* Reviewers point out that the method is very simple: place a colored marker on the query point in the first frame of the video and run the video generative model. This does not require any additional training of the model.
* Despite its simplicity, it outperforms other zero-shot tracking methods.


Concerns:
* Reviewers yFzS, duWk, sTu5, and nWQb highlighted the high cost of re-generating an entire video for every point tracked. The concern is that it will be difficult to use in real-world applications.
* Reviewers yFzS and duWk requested a comparison to DiffTrack paper, which also investigates emergent correspondences in diffusion models.
* Reviewer duWk argued that the tracking emerges from pixel-space heuristics (inpainting, color rebalancing) rather than the model's inherent ability to do the tracking.
* Reviewer yFzS questioned how the model handles occlusions or significant motion, as well as  robustness in diverse scenarios like poor weather or corrupted videos
* Reviewer duWk expressed a concern about the query point radius interacts versus input resolution and how this might bias comparisons across models.

**Reviewer Concerns:**

Concerns addressed by the authors:
* Runtime and cost: the authors conducted additional experiment to distill tracker into a small standalone feed-forward network that will be faster to run.
* DiffTrack paper: AC agrees with the authors that this work is concurrent with ICLR submissions and therefore the authors do not have to compare to it. In addition to this, authors showed that their method outperforms DiffTrack on the TAPVid-DAVIS dataset. Unlike DiffTrack, the current method is also architecture-agnostic.
* Handling Occlusions: authors clarified that the first stage (coarse propagation) regenerates the full video and captures large motions, while the refinement is purely local.


Remaining concerns:
* Long video tracking (reviewer yFzS). The authors have not tested the method on longer video generation that Wan2.1 supports
* Corrupted/poor quality video  (reviewer yFzS). The authors have not clarified the ability of the model to track points in less-than-ideal conditions.
* The dependency on the point radius (reviewer duWk)

**Reviewer Scores:**

The authors have addressed a shared concern about the cost, the purpose of the work and DiffTrack baseline. It is appropriate to raise the scores of the paper

Reviewer sTu5 - initial rating: 6  -> new score 7
Reviewer nWQb - initial rating: 6  -> new score 7
Reviewer duWk - initial rating: 2 -> new score 5
Reviewer yFzS - initial rating: 4 -> new score 6

---

### Decision · Program_Chairs · 2026-01-26

Accept (Poster)